# Technical Note: Differences in the diurnal pattern of soil respiration under adjacent *Miscanthus x giganteus* and barley crops reveal potential flaws in accepted sampling strategies.

J Ben Keane[1] and Phil Ineson[2].

[1]Environment Department, University of York, Wentworth Way, Heslington, YO10 5DD
[2]Department of Biology, University of York, Wentworth Way, Heslington, YO10 5DD

*Correspondence to*: Ben Keane: email ben.keane@york.ac.uk, telephone 01904 32344

**Abstract.** For convenience, measurements used to compare soil respiration ($R_s$) from different land uses, crops or management practices are often made between 09:00-16:00, convenience which is justified by an implicit assumption that $R_s$ is largely controlled by temperature. Three months' continuous data presented here show distinctly different diurnal patterns of $R_s$ between barley (*Hordeum vulgare*) and *Miscanthus x giganteus* (*Miscanthus*) grown on adjacent fields. Maximum $R_s$ in barley occurred during the afternoon and correlated with soil temperature, whereas in *Miscanthus* after an initial early evening decline, $R_s$ increased above the daily average during the night and in July maximum daily rates of $R_s$ were seen at 22.00 and was significantly correlated with earlier levels of solar radiation, probably due to delays in translocation of recent photosynthate. Since the time of the daily mean $R_s$ in *Miscanthus* occurred when $R_s$ in the barley was 40% greater than the daily mean, it is vital to select appropriate times to measure $R_s$ especially if only single daily measurements are to be made.

## Keywords

Soil respiration, *Miscanthus x giganteus*, barley, diurnal patterns, photosynthesis, carbon dioxide ($CO_2$), greenhouse gas (GHG), solar radiation, PAR

## 1 Introduction

Soil respiration ($R_s$) is a major process in the global carbon (C) cycle, contributing approximately 30% of ecosystem respiration (Bond-Lamberty and Thomson, 2010). Though the controls on $R_s$ are less-well described than for photosynthesis, as atmospheric carbon dioxide ($CO_2$) concentrations pass 400 ppm it is becoming increasingly important to improve our understanding of this important biological process. The implications that changes in $R_s$ might have for climate change have long been discussed (Schlesinger and Andrews, 2000) and in recent years the attention given to the potential of soils to sequester large amounts of carbon to mitigate rising levels of atmospheric $CO_2$ through management practices (e.g. Gattinger et al., 2012) demands that we measure all aspects of the global carbon cycle, including $R_s$, as accurately as possible.

The most common method used to measure $R_s$ is the closed chamber technique (Mosier, 1989) with manual chambers tending to be employed for sampling from a weekly to monthly basis (e.g. Drewer et al., 2012; Toma et al., 2011; von Arnold et al., 2005). $R_s$ is generally accepted to be largely controlled by soil temperature (Bond-Lamberty and Thomson, 2010) and if combined with an assumption that soil temperature will be consistent across a single site, a logical expectation

might be that the diurnal variation in $R_s$ will also be consistent at that site. Many studies consider it sufficient to use a single simultaneous daily measurement of $R_s$ to test for differences between different land uses or vegetation types and to extrapolate long-term budgets, (e.g. Barrena et al., 2013; Finocchiaro et al., 2014; Gauder et al., 2012; Johnson et al., 2010; Shvaleva et al., 2014; von Arnold et al., 2005; Zhang et al., 2013). Whilst the importance of selecting appropriate and synchronous sampling times is commonly recognised, measurement "windows" often vary across two hours (Kessavalou et

al., 1998; Zhang et al., 2013) to as much as seven (Finocchiaro et al., 2014) or even eight hours (Gao et al., 2014), generally between 09:00-16:00; however, none of these cited studies provided any data to support these windows which are largely based on minimising time delays between comparisons and assumptions that minimised temperature changes are the key to measurement parity. Although work has been undertaken to ascertain the most suitable time of day to sample $R_s$ manually (e.g. Wang et al., 2012;Savage and Davidson, 2003), these studies have focussed on a single vegetation type or land use, thus

do not resolve the issue of selecting the most appropriate sampling time at which to make comparisons between different experimental treatments or crops.

In the current work the aim was to compare the $R_s$ fluxes between two adjacent crops, as part of a fuller quantification of ecosystem C budgets. The two crops monitored in this study were the conventional arable crop barley (*Hordeum vulgare*), the second most widely planted arable crop in the UK (DEFRA, 2014), and the perennial grass species *Miscanthus x*

*giganteus* (henceforth *Miscanthus*), which is increasingly cultivated as an energy crop. In this study the use of automated chambers allowed the collection of near-continuous measurements of $R_s$ and the resulting data set was used to investigate the effect of sampling time and crop on $R_s$, and how this might differ across a period of several months.

## 2 Methods & materials

### 2.1 Study site and experimental design

Soil respiration ($R_s$) was measured using automated chambers and infrared gas analysers (IRGA, Licor LI-8100-101A, Lincoln NE, USA) with multiplexers (Electronic workshops, Department of Biology, University of York, York UK) beneath a seven year-old stand of *Miscanthus* and an April-sown spring barley in adjacent fields on a farm in the east of the United Kingdom, with one IRGA and one multiplexer deployed in each crop (see Drewer et al., (2012) for a full site description). Chambers (n=6) were placed at random within separate plots at least 1.5 m apart in the two fields and so were treated as

independent replicates; chambers were seated over PVC collars (diameter 20 cm, height 10 cm) which were inserted *ca.* 2 cm into the soil in order to minimise the effect of cutting fine roots (Heinemeyer et al., 2011) and these remained in situ throughout the study, which was undertaken from May to August 2013. The chambers were programmed to close for two

minutes during measurement, with a 30 second 'dead band' to allow for mixing of the headspace, in a continuous cycle between chambers. Collars did not exclude roots and no above-ground vegetation was included. Soil temperature and moisture at 5 cm depth were also measured every 15 minutes adjacent to each chamber collar and averaged over hourly intervals using vertically-installed sensors (Delta-T DL2 and GP1 loggers, SM200 soil moisture probes and ST1 temperature

probes; Delta-T, Cambridge UK), and hourly meteorological data (solar radiation, air temperature) were recorded onsite using a weather station (WP1, Delta-T, Cambridge UK).

## 2.2 Data processing and analyses

$R_s$ fluxes were calculated as linear regressions of $CO_2$ concentration against time and corrected for volume and temperature using the manufacturer's software (see manufacturer's manual https://www.licor.com/documents/jtpq4vg358reu4c8r4id.pdf)

and subsequent analyses were conducted using SAS 9.3 (SAS Institute, Cary NC USA). In the first instance the $R_s$ flux data were hourly averaged for each of the individual three months of the study, but to enable diurnal patterns to be more clearly identified, deviation from the daily mean was ascertained by subtracting hourly fluxes from the daily mean $R_s$ and the data for each month were subsequently averaged. Cumulative $R_s$ fluxes were calculated by trapezoidal integration for each chamber within both crops and averaged to estimate the total flux; data were not gap-filled, instead where there were gaps in

the data for one crop, the corresponding fluxes from the other were omitted from the calculation to estimate cumulative flux. This resulted in a loss of 15 days over the study period (five days in May, six in June and four in July) which represented a total coverage of 80%. These estimates were then used to investigate the influence of sampling hour on the monthly cumulative estimate of $R_s$ by comparing cumulative fluxes calculated using individual sampling hours (e.g. deriving a cumulative estimate of $R_s$ by integrating only fluxes measured between 14.00 and 15.00) and those using all measurements

for each month. The cumulative fluxes for the whole period were tested for normality using a Kolmogorov-Smirnov (K-S) test, but due to the size of the dataset this approach was unsuitable for the cumulative fluxes for sampling hour and instead limits of kurtosis and skewness of $\pm$ 2 were used as acceptable deviation from a normal distribution (Field, 2013). Differences in the whole-period cumulative flux were tested using one-way analysis of variance; the effect of crop, sampling hour and month were tested using a mixed-effects model accounting for the repeated estimated totals from each chamber for

each month (PROC MIXED in SAS, using the 'repeated' statement and an autoregressive covariance structure). Ancillary environmental data (soil temperature, soil moisture, solar radiation and air temperature) were averaged hourly and over each month using the same method applied to fluxes of $R_s$. These hourly averaged data were used in regression models to explain the diurnal pattern in $R_s$, and more detailed analyses were undertaken by performing separate regressions with flux measurements taken during the typical daily measurement window (09:00-16:00) and outside of this window. A further

analysis was completed by performing regressions of fluxes against 'lagged' measurements of solar radiation, i.e. the effect of prior levels of solar radiation on $R_s$ was tested.

## 3 Results and discussion

At the start of the study period (May) $R_s$ tended to be higher in the *Miscanthus* than the barley (Fig 1), but this reversed during June and higher fluxes of $R_s$ were consistently seen under the barley until the end of July. Highest rates of $R_s$ were seen in the barley during early July (*ca.* 1500 mg-$CO_2$ m$^{-2}$ h$^{-1}$) and declined soon after, whereas $R_s$ climbed steadily under the *Miscanthus* until it reached a maximum of *ca.* 800 mg-$CO_2$ m$^{-2}$ h$^{-1}$ towards the end of July (Fig 1).

The hourly monthly averaged fluxes revealed strong diurnal patterns for $R_s$ in both crops (Fig. 2). For all three months in barley, maximum $R_s$ was seen between 12:00-15:00, minimum around 05:00 and daily means at 09.00 and *ca.* 20.00. However, $R_s$ changed distinctly in the *Miscanthus* across the three months of the study. The magnitude of the daily variation in $R_s$ was remarkably different between the two crops (Fig. 2): for both barley and *Miscanthus* the daily minima were *ca.* 10 % below the daily mean across the study, but where the maxima in barley increased from *ca.* 15% in May, to 20% in June to as much as 40% above the daily mean in July, it declined in *Miscanthus* from 20% in May, through 15% in June and finally just 10% above the daily mean in July (Fig. 2). During May the daily pattern of $R_s$ was similar for *Miscanthus* and barley but in June, although $R_s$ peaked around 15:00, after initially declining it increased again so that for the period 20:00 to 04:00 was greater than the daily mean. This pattern for $R_s$ changed again through July, when the lowest daily $R_s$ was seen at 09:00 coinciding with the daily mean for barley, whilst $R_s$ for *Miscanthus* did not increase above the daily mean value until 18:00 peaking at 21:00, as much as five hours later than the peak in the barley.

The data did not significantly differ from a normal distribution (K-S test $D_{[10]}= 0.21$, $p> 0.05$; kurtosis= 0.25, skewness= 0.95). Cumulative $R_s$ flux was greater from barley over the entire study period ($F_{[1,8]}=6.62$, $p<0.04$), there was a strong and significant effect of the chosen sampling hour on that estimate ($F_{[23,568]}= 4.28$, $p< 0.0001$) and a resulting strong significant difference between monthly totals ($F_{[2, 568]}= 901.35$, $p< 0.0001$). There was a significant interaction between sampling hour and crop type ($F_{[23,568]}= 3.40$, $p< 0.0001$), and a further significant interaction between crop and month ($F_{[2,568]}= 202.44$, $p< 0.0001$), emphasising that it is not at all valid to assume that measurements made in the adjacent two crops at the same time were sufficient for comparisons of total $R_s$ flux.

Questions must be raised regarding the validity of using blanket, common sampling strategies to compare $R_s$ between different vegetation types, given the marked diurnal changes in $R_s$ demonstrated here. Indeed, if a protocol were employed which used the same sampling hour over several months, the significant interaction between crop and month shows that the shift from higher $R_s$ in the *Miscanthus* in May to higher fluxes from the barley in June and July would be totally missed. For example, considering only the measurements taken around 15:00 in this study, in May not only would the cumulative $R_s$ from both crops be overestimated, it would be concluded that $R_s$ from barley was higher than or the same as for *Miscanthus*, when that clearly is far from correct (Fig. 3). Over the entire study, measurements made singly at just 15:00 would further bias the conclusions, so that in July $R_s$ from the barley would be overestimated by 40%, whilst there would be a slight underestimate from the *Miscanthus*, introducing the real possibility of not only exaggerating differences between crops, but also of creating artefactual differences simply resulting from the choice of a standardised measurement protocol.

Analysis of environmental variables showed that $R_s$ in the barley was a function of soil temperature (Fig. 4). Soil temperature also had a strong positive effect on $R_s$ (Fig. 4) in the *Miscanthus* between 09:00-16:00 but it did not explain the night-time fluxes. during which time $R_s$ was strongly positively correlated with the level of solar radiation seen earlier in the day (Fig. 5). Several studies have ascribed such hysteresis or apparent asynchronous $R_s$ response to soil temperature to a discrepancy between depth of $R_s$ source and the measurement depth of soil temperature (e.g. Oikawa et al., 2014; Graf et al., 2008; Pavelka et al., 2007) and this explanation cannot be discounted for the response seen here in *Miscanthus* since this study is limited by soil temperature measurements at a single depth (5 cm). Soil moisture has also been proposed as the driver of temperature hysteresis (Ruehr et al., 2010; Riveros-Iregui et al., 2007), though our analysis did not find that relationship on a diurnal scale: multiple regression of $R_s$ with soil temperature and soil moisture did not improve the explanation of the daily variation in $R_s$. There was a short period (19th – 22nd July) however, following two weeks without rain, when soil moisture dropped to a low of 0.16 $m^3$ $m^{-3}$ in the arable crop and during this time $R_s$ dropped considerably (Fig. 1). When heavy rainfall elevated soil moisture, rates of $R_s$ increased again which would suggest there is a threshold above which soil moisture is not limiting, an effect similar to that described by Xu and Qi (2001).

Alternatively, if solar radiation is considered a proxy measurement of photosynthesis, the delay in response of $R_s$ may be a function of photosynthate translocation to roots and the rhizosphere, which has been shown to be important to all component processes of $R_s$ (e.g. Heinemeyer et al., 2012) and having witnessed such a lag in an oak savannah system, Baldocchi et al. (2006) propose a similar explanation. This is further supported by Gavrichkova and Kuzyakov (2008) who showed that under constant temperature a diurnal response in $R_s$ will still be evident under maize (*Zea mays*) but not from unplanted controls, and another study which demonstrated that shading maize plants will reduce the diurnal pattern in $R_s$ (Kuzyakov and Cheng, 2004). This suggestion is further strengthened as the delay observed in the current study increased as the *Miscanthus* crop grew taller; from six hours in May, to seven in June and ten in July. It is known that translocation is slower in taller vegetation and may also be slowed as transpiration increases (Kuzyakov and Gavrichkova, 2010), as would be expected later in the summer. An obvious physical difference between the two crops monitored in this study is that of size, with *Miscanthus* exceeding 3 m when fully grown and barley less than 0.5 m, so the speed of translocation in barley may be quicker and therefore the effect of photosynthesis in this crop is more confounded with soil temperature (Kuzyakov and Gavrichkova, 2010). Differences in the diurnal pattern of $R_s$ have been demonstrated between grass species and mesquite trees in savannah ecosystems (Barron-Gafford et al., 2011), and again between grasses and forest soils (Heinemeyer et al., 2011) which both reflect the differences presented here of temperature decoupled peak in $R_s$ under the taller trees occurring later in the day. Such a lag in $R_s$ cannot be assumed under all tall vegetation however, as studies under maize and switchgrass (*Panicum virgatum*), which share the physiological traits of height and C4 photosynthesis with *Miscanthus*, demonstrated a clear diurnal relationship between $R_s$ and soil temperature (Han et al., 2008; Huang et al., 2016).

A lack of consensus persists regarding the cause of these lags in $R_s$, a point acknowledged by Phillips et al. (2011) in a study which used computer modelling to attempt to interpret hysteresis, and their analysis led them to conclude that the phenomenon might possibly be due solely to physical, not biological processes. A more recent modelling study provided

further explanation of how both photosynthate and soil moisture might affect observed hystereses (Zhang et al., 2015). On the balance of our analysis and the literature cited here, we are inclined to hypothesise that it is the former which drives the lag presented in our data. However, it should be reiterated that a definitive explanation of the drivers of $R_s$ hysteresis was beyond the scope of the current study and further targeted experimental work should be implemented if this additional aim is to be achieved.

## 4 Conclusions

In this study strong, clear diurnal patterns in $R_s$ have been demonstrated, and these are not consistent between different crops, even at a single location. Without the use of an automated flux measurement system, this discrepancy would not have been identified, although it is acknowledged that manual sampling techniques have an important role to play particularly when cost of equipment and access to power are a common limitation. It is therefore a matter of great importance that sampling strategies founded upon single daily measurements of $R_s$ are undertaken at a time representative of the daily mean flux, and in order to do so it is absolutely vital that a thorough understanding of the diurnal variation is used to guide any sampling strategy. It is therefore suggested that especially in manual sampling experimental designs, the diurnal pattern of $R_s$ is first established by measuring across a full 24 hour cycle and that this is revised periodically, since it has been shown here that the diurnal cycle may change greatly over several months. Failure to do so may lead to inaccurate long term estimates, and in experimental contrasts it may cause grossly incorrect (by as much as 40% relative to the respective daily means) conclusions to be drawn. Since $R_s$ is such a critical component of the global carbon cycle, it is essential that our understanding of this process, and how it is effected by management practices, be founded upon accurate data, which will only be achieved through well planned sampling strategies.

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

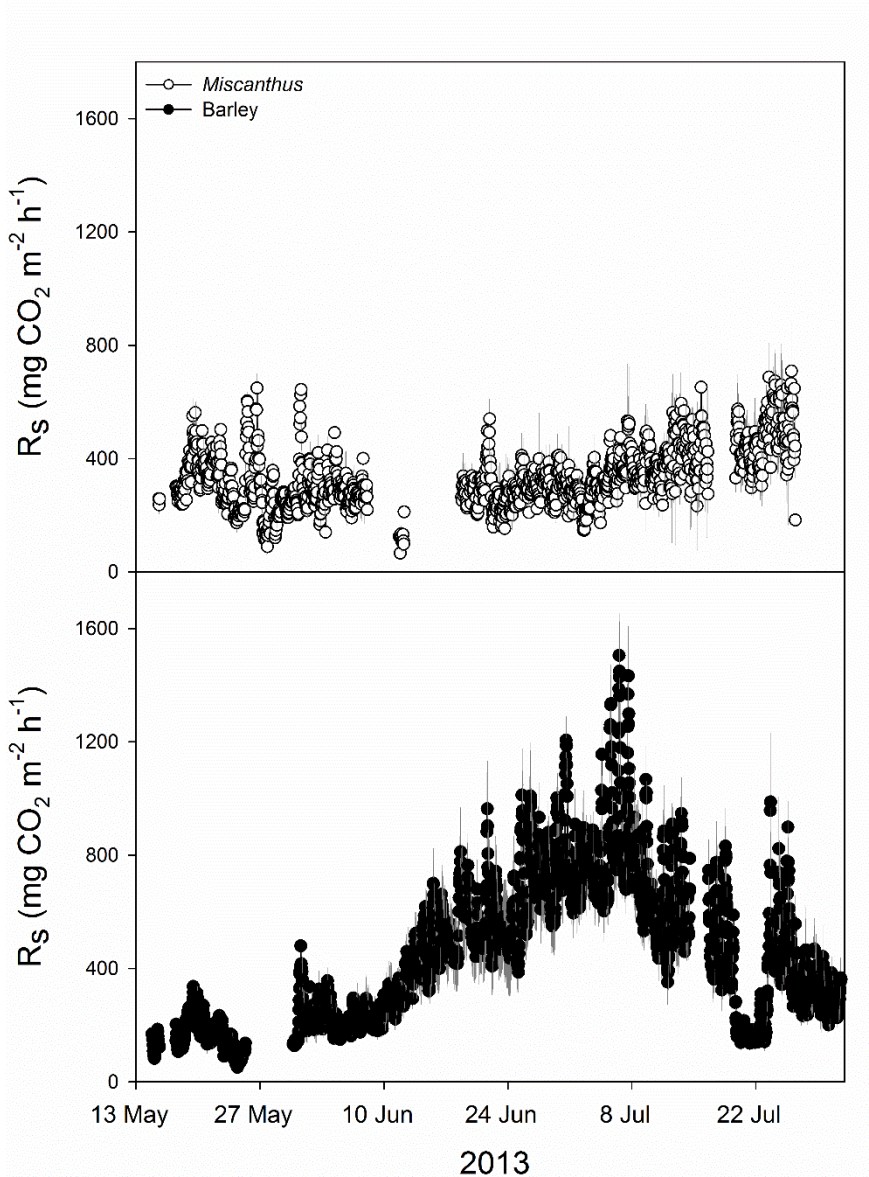

**Figure 1.** Mean (± 1SE, n=6) $R_s$ from under *Miscanthus* (top panel) and barley crops (bottom panel) during summer 2013, measured using Licor automatic flux chambers.

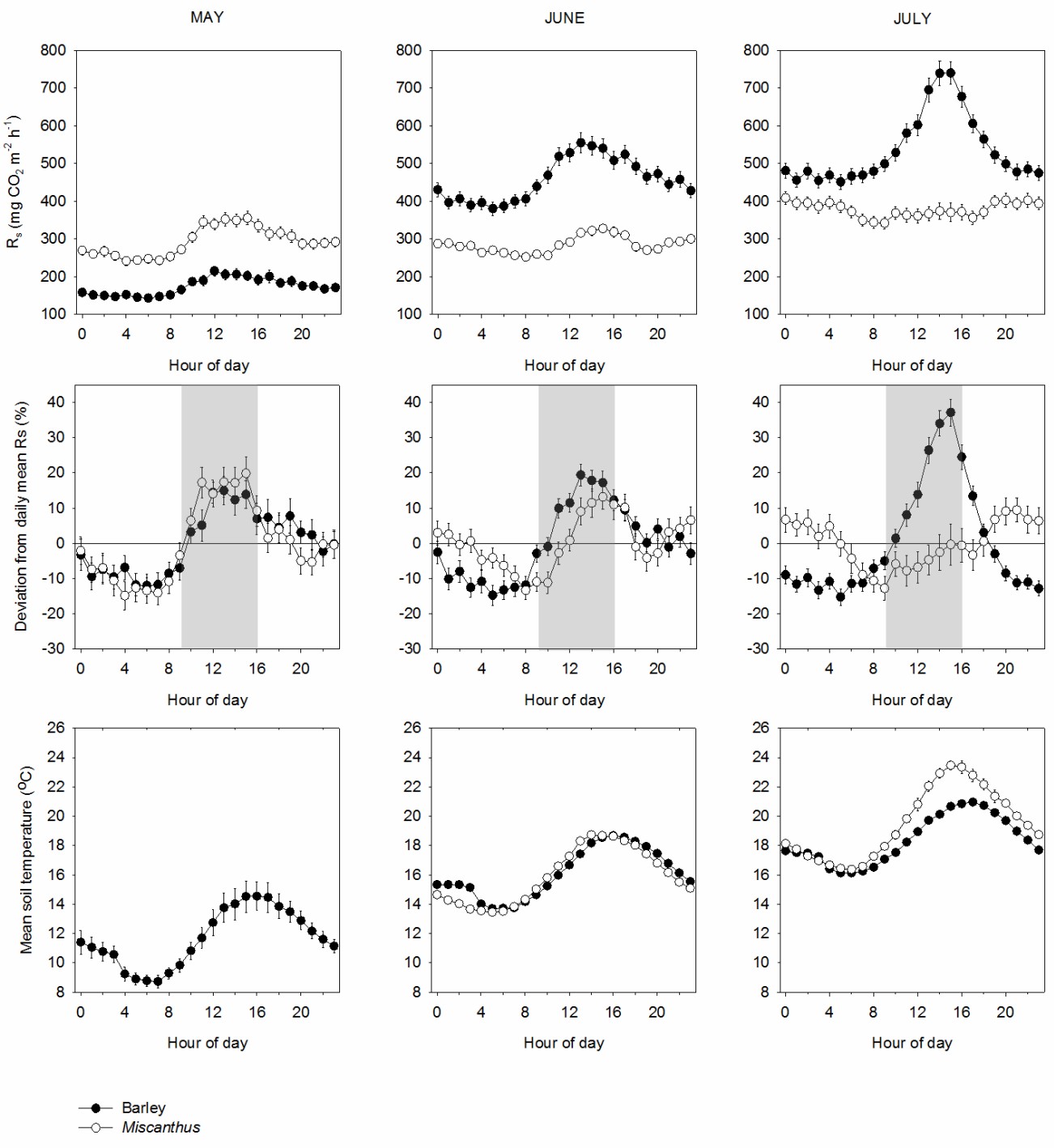

**Figure 2.** The diurnal pattern of $R_s$ and soil temperature at 5 cm depth for each month of the study for barley and *Miscanthus* crops. Values shown are mean ($\pm$ 1SE) average hourly absolute values of flux $R_s$ (top row) and deviation from the daily mean (middle row). The shaded area of the middle panels represents the typical measurement window during which manual

sampling would take place. Zero deviation represents the daily mean flux, positive deviation representing fluxes greater than the mean and negative fluxes smaller than the mean.

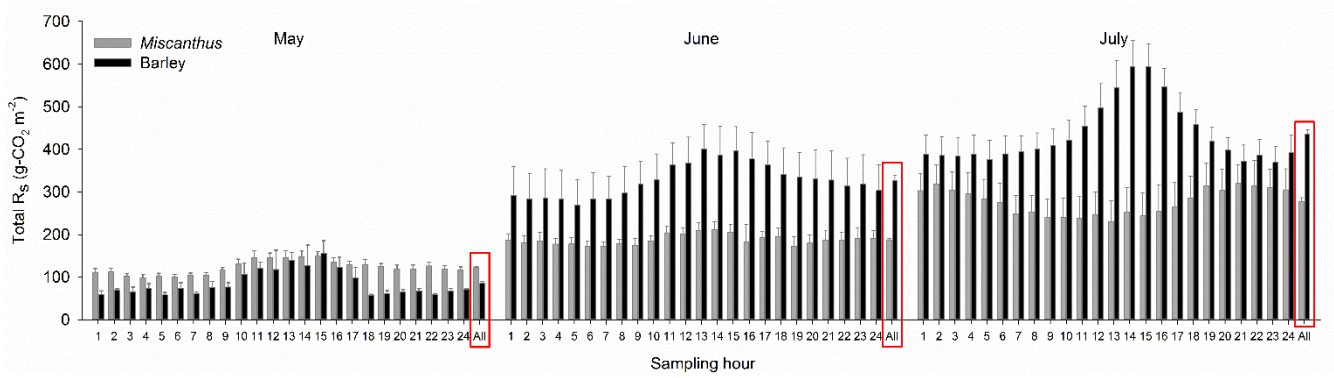

**Figure 3.** Estimates of the cumulative flux $R_s$ under *Miscanthus* and barley crops using measurements taken using only single hours (1-24) or continuous measurements (All) across three months in summer 2013. Values shown are mean cumulative flux (± 1SE, n=6).

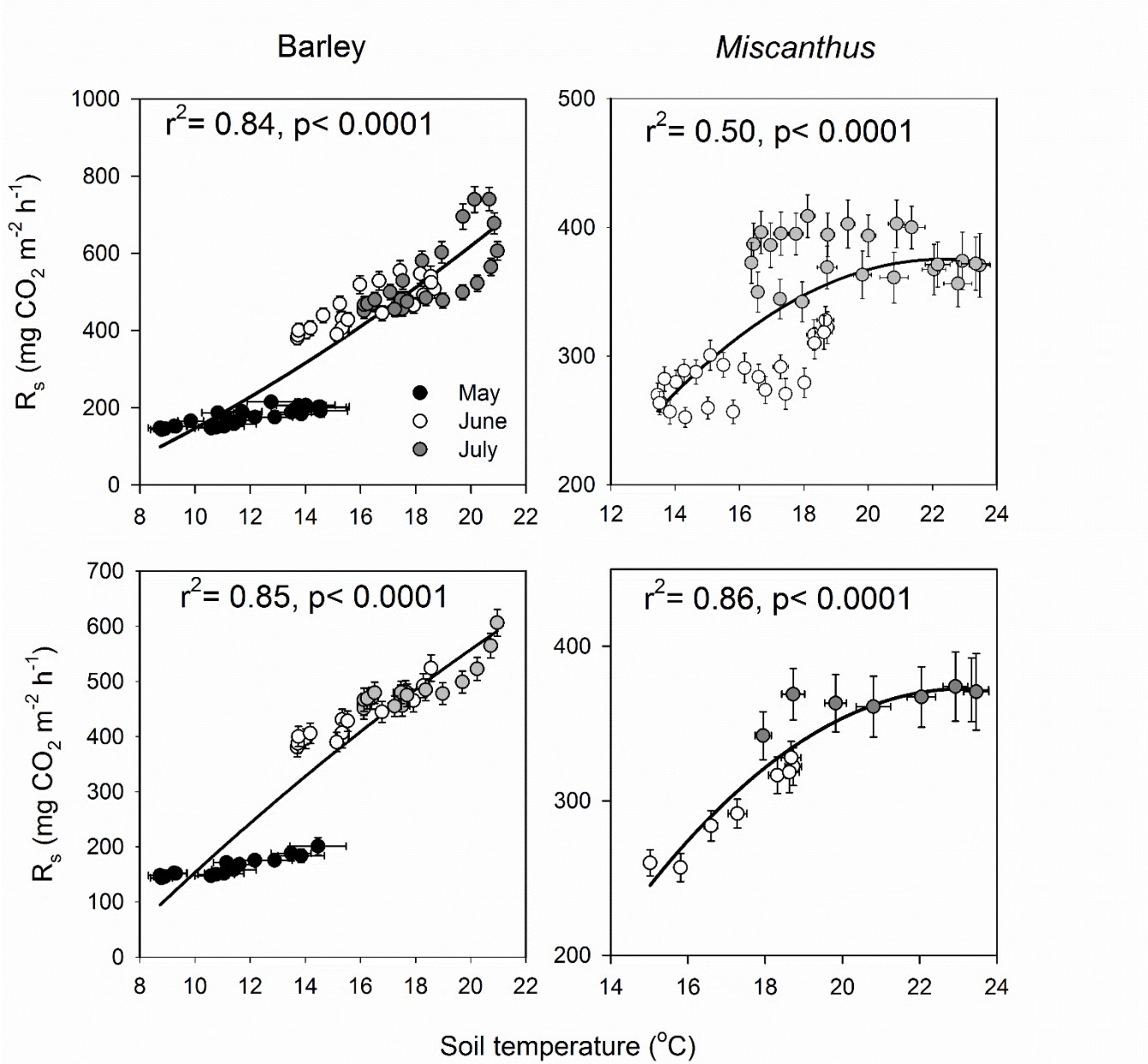

**Figure 4.** Regression models of monthly mean average hourly (± 1SE, n=6) flux $R_s$ and soil temperature at 5 cm depth for barley (left column) and *Miscanthus* (right column). Data shown include full 24 hour period (top row) and only data from the typical manual measurement window of 09:00- 16:00 (bottom row). Soil temperature data were not available for *Miscanthus* during May.

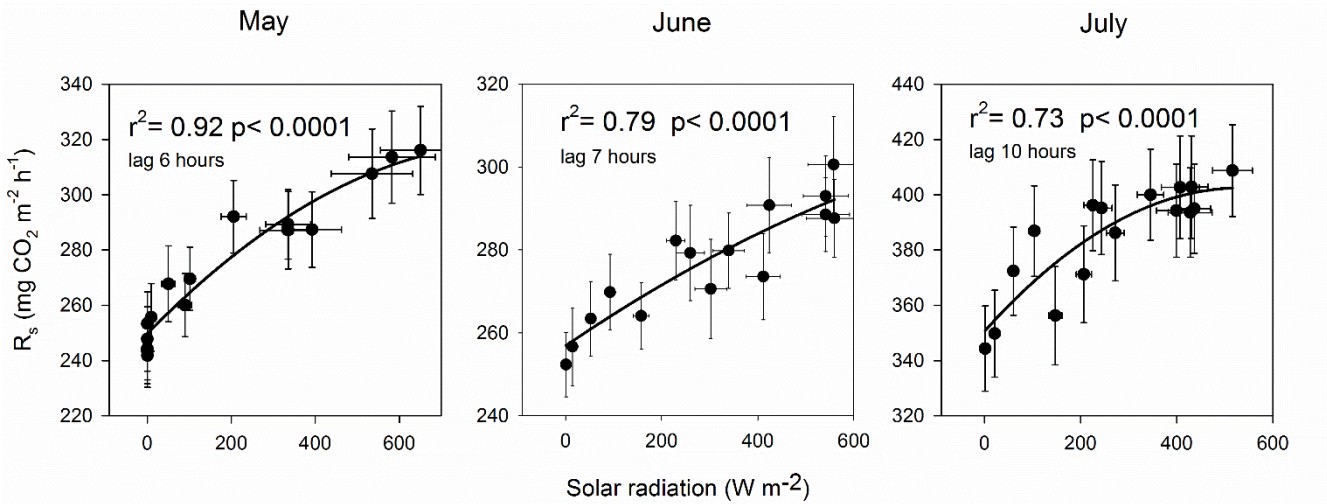

**Figure 5.** Response of $R_s$ to preceding levels of solar radiation in *Miscanthus* outside of the typical manual measurement window (see text). Values shown are hourly means (±1SE n=6) averaged over each month. The lag time is the length of the offset between the measured solar radiation and the $R_s$, e.g. for May the relationship shown is that of solar radiation at 12.00 and $R_s$ measured at 18.00 (lag time= 6 hours) and the lag times shown for each month are those which yield the closest relationship (highest $R^2$).

**Acknowledgements**

This work was funded by the Energy Technology Institute as part of the ELUM (Ecosystem Land Use carbon flux Modelling) project. The authors thank the landowner Jonathan Wright for access to the site and to Steven Howarth and Trevor Illingworth of the Biology Department Electronic Workshop for providing extensive electronic engineering support, and Sylvia Toet for providing advice on the manuscript.

