# Peer review of "Technical Note: Differences in the diurnal pattern of soil respiration under adjacent *Miscanthus x giganteus* and barley crops reveal potential flaws in accepted sampling strategies."

_Biogeosciences, 2016_

## Short Comment (SC1) · 28 Oct 2016

With great interest, I was looking for the differences of soil respiration under the two crops. However, it was not clear. Respiration may correlate to T but also water or carbon availability. Aside from methodological considereration, it would be interesting to clarify in the abstract or even in the title, if appropriate "Miscanthus induced lower soil repiration that barley under our temperate climatic conditions", "The seasonal and daily pattern was ..." The MS should focus on significant findings and the conclusions

should be drawn including ecological findings - for BIOGEOSCIENCES readers.

---

## Author Comment (AC1) · 28 Oct 2016

I will be happy to amend the abstract to include more detail regarding the magnitude of fluxes between the two crops.

---

## Referee Comment (RC1) · J.W. Atkins (Referee) · 4 Nov 2016

Review of Technical Note: Differences in the diurnal pattern of soil respiration under adjacent Miscanthus x giganteus and barley crops reveal potential flaws in accepted sampling strategies. Biogeosciences Discuss., doi:10.5194/bg-2016-397, 2016

from Jeff Atkins (jeffatkins@virginia.edu)

I think this is an excellent contribution to an important experimental design and sam-

pling issue that has been a concern in some circles of the carbon cycling community for a while. Showing differences in diurnality between these two vegetation types is an important consideration—and that the diurnality changes seasonally as well! This work will be of interest to the Biogeosciences community. I think this work, though it is a short technical note, could be improved further.

In the abstract, there is discussion of correlations with solar radiation and later in the paper discussion of soil moisture effects, though these are not quantitatively included later or specifically addressed. Nor are site effects such as soil carbon, soil nitrogen, etc. While a full description of the site is referenced, a brief inclusion of a sentence here to help understand site differences or homogeneity would be beneficial given the implications for carbon cycling. There is also a body of work on soil carbon flux hysteresis that is not included or referenced. I have made some notes of that later in this review. I offer this as it would add to this work, provides sound theoretical background, and will also help better frame how this work can aid researchers in the future. I have noted some considerations for the figures as well.

Pg 1, line 26 need an "of" between "amounts" and "carbon"

Pg 2, line 4 – this sentence is worded awkwardly and could be focused more. Monitoring would imply fixed-chambers, but many studies employ portable chambers—some of which require in-situ collars and some that don't.

Pg 2, line 13 – "these studies have focussed on a single vegetation type or land use thus do not resolve . . ." Focused has an "s" too many and you need a conjunction between "use" and "thus."

Pg 2, lines 10-20 – There is some work from Diego Riveros-Iregui that would be a valuable contribution here about diurnal hysteresis—if not in this section to set the scene, perhaps later:

Riveros‐Iregui, D. A., Emanuel, R. E., Muth, D. J., McGlynn, B. L., Epstein, H. E.,

[Figure]

Welsch, D. L., ... & Wraith, J. M. (2007). Diurnal hysteresis between soil $CO_2$ and soil temperature is controlled by soil water content. Geophysical Research Letters, 34(17).

And this paper may also be useful:

Ruehr, N. K., Knohl, A., & Buchmann, N. (2010). Environmental variables controlling soil respiration on diurnal, seasonal and annual time-scales in a mixed mountain forest in Switzerland. Biogeochemistry, 98(1-3), 153-170.

Pg. 3, line 30 – just a note on units grams per hour or micromoles per second are typically more common in the literature.

Pg. 4, lines 1-3 – I don't completely understand what you are saying with this phrase, " . . . and daily means at 09.00 and ca. 20.00 for all three months in barley" could you clarify that? Are you saying that is when fluxes approximate daily means?

And a really, really minor point, but I think "greater" works better to describe fluxes than "higher" because you are talking about a magnitude, an accumulating sum of sorts.

Pg. 3, lines 16-20 – Love it. That is a great point and I am enthused to see this work on experimental design and sampling! That is a good highlight to show that difference and make that point about missing differences between the systems.

Pg. 4. – It would be helpful if you showed your soil moisture data or described it in some way and provide analysis of how that is working with temperature or in isolation to control fluxes. That interaction can be important. There are various ways to look at the interaction of temperature and moisture such as an ANCOVA or even looking at some log regression detrending. Inclusion of an ANCOVA would likely address this and be of minimal additional work.

Fig. 1 – there is a bit of an over-plotting issue with the data that could be addressed by perhaps widening the plot or decreasing the marker size.

Fig. 2 – Great plot in general, but I think that changing the scale on the y-axes, though

I understand visually why it was done, is not a good practice. Normalizing those axes would also better show monthly differences—as you can see in the soil temp. plots at the bottom.

---

## Referee Comment (RC2) · Anonymous Referee #1 · 9 Nov 2016

General comments: This is an interesting study. The fact that it is submitted as a technical note is maybe confusing, as it attempts to resolve plant/soil physiological phenomena at the same time as making a point regarding measuring techniques. As far as the technical point is concerned, I think the authors argue very well that the diurnal fluctuation of respiration (Rs) has to be known for a given setting in order to put single measurements into context. I have oly some technical comments on that score (see detailed comments below).

[Figure]

Where the physiology of respiration is concerned, I think that more care is needed in how data are presented and interpreted. Based on the objective to assess Rs measurements during a specific daytime window, some data are separated into those obtained between 9:00 and 16:00 when drivers of Rs are considered. This splitting of data is however entirely arbitrary as far as regressions with temperature or lagged radiation is concerned. Continuous data sets are obviously powerful in resolving temporal dynamics, but rather than presenting a complete time series analysis, the authors split fluxes into convenient sections to assess drivers independently, when clearly photosynthate supply dynamics as well as temperature effects both act continuously. What I can see in the presented data are hints at possible flux dynamics based on diurnal photosynthate supply, but this accounts for a rather small variation of mean daily fluxes. Fig. 2 emphasises deviations by using % of flux mean, but these are rather small in absolute terms.

Also the temperature dependence is treated somewhat inconsistently. It was not clear to me why the temperature response was plotted for monthly average values by hour, rather than the raw data or daily averages. Averaging by hour integrates values over up to 30 days, during which time considerable variation can occur. What's not at all clear is why all diurnal data are presented for one data set (barley), but only a subset for the other (Miscanthus). If the authors wanted to show that there is a distinction between different time of day, then a much better way would be to include those data too and use different colour/shading to make this point.

Specific comments: P. 1, l. 8-10: I don't follow the logic of the sentence. The first half refers to a mode of measurement, based on "convenience" of working in daylight hours, the second invokes an assumption that temperature is a dominant control of soil CO2 efflux. Why this conflation? I assume you want to set up the issue of contrasting diurnal maxima periods, but this is not at all clear in the way it is phrased.

P. 1, l. 12: The statement that Rs in Miscanthus peaked in the night is not true. For May and June, Miscanthus "peaks" during the 9:00 – 16:00 window. Diurnal variations

in July are subtle (+/- 10%), and interpretation should take account of this magnitude.

P. 2, l. 3-7: Here is the same conflation of measurement mode and temperature control. The two concepts are not logically linked here – the single measurement is not a consequence of temperature being widely held as a dominant control on respiration, as the sentence suggests.

p. 3, l. 30: As you chose to express fluxes on a mass basis, please specify whether these are grams of carbon, or grams of CO2. Using molar units would avoid any confusion.

P. 4, l. 12: "fully"???

P. 4, l. 17: Which protocol do you refer to here?

P. 4, l. 30-32: Picking-and-choosing your data points so they fit the narrative is not appropriate. The temperature response for both data series have to be balanced, and you should show all hourly data for Miscanthus in Fig. 4. Or, as you are interested in a temperature regression across all months, I'm not sure that hourly data are meaningful to show in any case. It's a shame that you don't have temperature data for June in Miscanthus, but for a seasonal temperature response (which is what is sown by regression lines in Fig. 4), you can use monthly average Rs and soil temperature measured in barley as an approximation. From Fig. 2, this would place fluxes of around 300 mg CO2 m-2 h-1 near 12 deg C – what does that do to your curve? Regarding your regression functions – is an apparent saturation curve for the temperature response meaningful for Miscanthus? Finally, there seems to be a mismatch between short-term temperature response (e.g. June in barley, where diurnal flux response to temperature change is very sluggish) vs. seasonal response – this may be worth commenting on.

P. 5, l. 5: delete "however" (not needed as you start the sentence with "although")

P. 4, l. 32 – p. 5, l. 2: I don't completely follow this analysis. Why do you suppose that the relationship between solar radiation and soil CO2 flux is linked to the "typical

measuring window"? It seems an entirely arbitrary separation of daytime/nighttime of your data set. What I can see in these graphs is that by introducing a time shift between two essentially sinusoidal curves, you can create an apparent correlation. . The same analysis would work for soil temperature with a time lag, but I obviously see what you're getting at with the lag analysis. An analysis of regression between instantaneous flux and preceding photosynthesis (or radiation used as a proxy) would be more meaningful. If what you try to show is the case, then the deviation from the mean in CO2 flux should be greater during nights following days with high radiation, and less following days of low photosynthesis (i.e. low radiation).

Figure 1: Placing both data series on top of each other is not helpful. Please split into separate panels. What happened around the 20th July in the barley field? It seems strange that fluxes should suddenly fall dramatically and then remain constant for days (with only little diurnal variation visible), to then jump back. Any hints in the meteorological data or management (harvest)? What is the impact on your diurnal calculations?

Figure 3: I'm not sure that this graph provides much new information It should be the same as Fig. 2, only that average fluxes per hour and month are multiplied by the number of measurement days, or not? Dynamics should hence be identical.

---

## Short Comment (SC2) · 10 Nov 2016

Good work and important contribution to a growing evidence regarding using manual and thus time constraint soil respiration data to derive modeled (upscaled) estimates for longer time periods (months, years).

Please consider previous related work by Savage & Davidson (2003) & Xu & Qi (2001) as cited in the diurnal soil respiration trials done by Heinemeyer et al. in EJSS: one

on peatland, forests and grassland (2011) testing exactly this manual time constraint and in relation to root vs decomposition flux components, and one other study on component fluxes in a grassland (2012), the latter clearly showing a link to PAR (and thus GPP) for both, root and mycorrhizal flux components. I am somewhat surprised both these citations (alongside those key ones cited in the paper) and the relevant discussion sections addressing the potential implications for modelling and upscaled estimates have been omitted here.

---

## Short Comment (SC3) · 10 Nov 2016

It would be good in the context of this paper (which will be read by people hopefully avoiding sampling mistakes) to cite the collar depth paper by Heinemeyer et al. (2011), clearly showing the implications on overall soil fluxes and the component fluxes. This is also of importance for crops, particularly when long lived crops are under investigation.

It would certainly be good to cite the paper in the M&M section where it is indirectly
refered to (avoiding cutting roots!). And possibly in the discussion section as lost root fluxes will impact on any observed (or not) time-lag responses (ie if roots have been cut, prevented from entering the monitoring area, then a time lag will be less. This will be particularly the case for shallow rooted vegetation, in this case barley. Already a few centimeters will make a massive difference as nearly all fine roots are located in the top 5 cm. So both, sampling time together with collar insertion (if used) are important to consider when wanting to upscale measured fluxes and capturing time-lag effects.

---

## Short Comment (SC4) · 13 Nov 2016

The paper is a nice work for guiding soil respiration measurement design.

Since the temperature response of soil respiration is so important to your topic, I assume the widely reported soil respiration-temperature hysteresis should be addressed. And you did discuss a little in 3.2 Environmental control of Rs, however, I think this section could be discussed even better by incorporating the knowledge from a few previous

efforts. Please see follows.

For the diurnal scale soil respiration-temperature hysteresis, there are a few representative works, including the classic Phillips et al. (2011) paper that applies mathematical models answering a few fundamental questions, like how soil temperature measurement depth selection, heat flow influence the respiration-temperature relation, etc; Afterwards, Zhang et al. (2015) combined both model exercise and field experiments to give a more comprehensive explanation of the occurrence and mechanism of the hysteresis. To exclude the possible effect of temperature depth selection by plotting respiration and temperature colocated at the same depth, this work demonstrated how heat flow, gas diffusion, photosynthesis contribute to the hysteresis, and also explained how soil moisture modulates hysteresis magnitude. Actually, the hysteresis may be more widely reported than the authors realized, see the literature list that reported field measured soil respiration-temperature hysteresis in Zhang et al. (2015).

As a useful knowledge to this manuscript, the argument that "Even the CO2 flux (F(z)) and the environmental conditions at the same depth can be out of phase, since the flux integrates sources from other depths, causing hysteretic loops" (Zhang et al., 2015) would help explain why the temperature-depth selection cannot avoid hysteresis.

Another useful information for this manuscript is related to photosynthesis control on soil respiration. As photosynthesis has long been suggested as the determinant of soil respiration by providing respiration substrate (e.g., Kuzykov and Cheng, 2001; Kuzyakov and Gavrichkova, 2010), Zhang et al. (2015) suggested the time-delayed photosynthesis impact on soil respiration contribute to the '8' shaped soil respiration-temperature hysteresis, and altered the hysteresis direction (clockwise cycle, or counterclockwise cycle) under different time lag levels of transferring photosynthate from leaves to roots. But these are numerical modeling representations, Zhang et al. (2015) also acknowledge more field validation are still required. The authors can think a little about this.

A third representative and nice work is by Oikawa et al. (2014) as you cited.

references:

Kuzyakov, Y., and W. Cheng (2001), Photosynthesis controls of rhizosphere respiration and organic matter decomposition, Soil Biol. Biochem., 33(14), 1915–1925.

Kuzyakov, Y., and O. Gavrichkova (2010), REVIEW: Time lag between photosynthesis and carbon dioxide efflux from soil: a review of mechanisms and controls, Global Change Biology, 16, 3386-3406.

Oikawa, P. Y., D. A. Grantz, A. Chatterjee, J. E. Eberwein, L. A. Allsman, and G. D. Jenerette (2014), Unifying soil respiration pulses, inhibition, and temperature hysteresis through dynamics of labile soil carbon and O2, J. Geophys. Res. Biogeosci., 119, 521–536.

Phillips, C. L., N. Nickerson, D. Risk, and B. J. Bond (2011), Interpreting diel hysteresis between soil respiration and temperature, Global Change Biol., 17, 515–527.

Zhang, Q., G. G. Katul, R. Oren, E. Daly, S. Manzoni, and D. W. Yang (2015), The hysteresis response of soil CO2 concentration and soil respiration to soil temperature, Journal of Geophysical Research-Biogeosciences, 120, 1605-1618.

---

## Short Comment (SC5) · 17 Nov 2016

We welcome the positive comments regarding the ms. Thank you for bringing these papers to our attention, Heinemeyer et al (2011) is of particular interest due to the diurnal variation across three sites which complements the current study making the comparison at a single site; Xu & Qi (2001) is a valuable reference and Heinemeyer et al. (2012) provides useful insight to soil CO2 efflux and supports our assertion of the importance of PAR- we will make the relevant citations in the revised ms. The Savage

and Davidson (2003) paper is currently cited in the ms.

---

## Author Comment (AC2) · 17 Nov 2016

Our method does indirectly refer to the paper Heinemeyer et al. (2012) and we will make the citation in M & M. It also is relevant to our discussion section and will be cited appropriately in the revised ms.

---

## Referee Comment (RC3) · Anonymous Referee #4 · 20 Dec 2016

Review on "Technical Note: Differences in the diurnal pattern of soil respiration under adjacent Miscanthus x giganteus and barley crops reveal potential flaws in accepted sampling strategies"

The abovementioned manuscript presents sound evidence that repeated manual field measurements of soil respiration at an arbitrarily chosen fixed time of day does not only likely fail to reproduce long-term averages of this important flux (which is probably not in conflict with any serious claim in existing literature), but even can cause

significant errors in a comparison of two nearby treatments (where it is a widely accepted compromise). This demonstration is surely worth publishing in BG, although SC2 indicates that this might not be the first such study, and care should be taken to acknowledge prior ones appropriately. The manuscript is well written and mostly clear, in some points pointed out below, information e.g. on the methodology is missing or too scarce, and the present shortness of the manuscript offers a good opportunity to improve this while still staying concise.

The maybe most important (and only major) flaw is that the two topics presented in sections 3.1 and 3.2 are little connected and differ strongly in terms of originality, soundness of the methods applied, and speculativeness of the discussion. Topic 3.1 is clearly at the heart of the study, as indicated by the title, and the reason why I believe this manuscript should be published in BG. Concerning topic 3.2, I am unsure how much the authors can do with the measured data at hand to improve it up to a point where it can amend existing knowledge, so in my opinion two options - extending it for the analyses and discussions suggested by earlier reviewers and also below, but also cutting it down to what can actually be said, and spending more space on a more elaborate analysis and discussion of the main topic of the study, could both be considered.

Abstract

p1L14: "coincided with levels" - unclear, reword (see also comment on p5L32).

2.1

This section in general: How often and for how long were the chambers closed?

p2L23: ...and *an* infrared gas analyser? p2L24: specify: was it 2 multiplexers (one per ecosystem?) p2L28: inserted 2 cm: It is not mentioned which collar height was chosen (Li-Cor's standard?) and/or how high they protruded above the soil surface. In general, an insertion depth of 2 cm is rather low (possible lateral diffusion in coarse soils) and the resulting large height above the surface should be avoided because of

its altering effect on insolation, precipitation and wind (probably not so much an issue once the plant canopy is closed). p2L30: Give more details on sensor installation (vertical or through a trench, resulting depth averaging). Note that to gain confidence in the later discussion on (partly lagged) responses to temperature and solar radiation, the temperature would ideally have been measured in several depths.

2.2

p3L6: Licor software and manual sounds a bit odd, maybe "manufacturer"? p3L14: duplicate dot after 80% p3L17-20: Try to secure the reproducability of the statistical methods not so much (or at least not only) by telling which option of the applied software was chosen, but rather by referring to the name of the test, to literature if necessary, etc., e.g. which test for normality? The result on normality does not seem to be mentioned in the results section (if I didn't overlook it). Note that for soil respiration in general it wouldn't be surprising if it was lognormal rather than normal, where necessary some authors work with log-transformed values.

3.1

p4L3: 9:00 and 20:00: unclear, you mean that instantaneous values close to the daily mean were reached at these times of the day? Reword. p4L18: "...shows that the shift [...] would be totally missed": This type of very straight conclusion would better fit in the following paragraph, where such things are plainly demonstrated.

3.2

p5L1/Fig.5: Make clear that the lag shown in the figure for each months is the one that yielded the optimal R2 after experimentally testing all lag times in a range from x to y in steps of z (here and/or near p3L25 in 2.2).

p5L3-7: Although this hypothesis is plausible for your case, little is presented to support or falsify it. If radiation data are experimentally shifted to improve $R^2$, so should be temperature data to check for the effect of the mentioned lagged response by improper

temperature measurement depth (ideally it would have been measured at more than 1 depth, see comment on p2L30). The physically most consistent way to do so would be by Fourier analysis, since heat transport in the soil would introduce different delays for temperature variations on different temporal scales (e.g. diurnal cycle vs. slower or faster variations), but if variability in a certain time-window is strongly dominated by the diurnal cycle, a simple shifting might do as well. Also, the sentence is very long. Its 2nd half is unclear to me and should be reworded. It seems that a single case study, where hysteresis in the Rs-T relation could be attributed to photosynthates after comprehensive measurements, is used to infer that the same is true in your case. At the same time, an abundance of literature is ignored which demonstrates that also heat transport and measurement depth effects alone can cause hysteresis (e.g. Pavelka et al., 2007, Plant Soil 292:171 and Graf et al., 2008, Biogeosciences 5:1175 to mention just the earliest systematic studies, many follow-ups have been already mentioned by other reviewers).

p5L32: Specify what exactly (e.g. the ratio or difference in total repiration between two treatments) can be incorrect by 40 % - the way it is written now suggests that conclusions are, but what would be a 40 % incorrect conclusion?

———————————————————

---

## Author Response (AR1)

**Manuscript bg-2016-397**

**Authors' response to reviewers' comments**

**General comments**

We thank all of our reviewers and our pleased to see that our manuscript has been positively received. We have already

5   addressed short comments SC1-SC3 and here we respond to the outstanding short comment SC4 and the reviewer comments RC1-RC3.

Our reviewers provided some insightful criticisms and several relevant additional sources which we have incorporated into the revised manuscript. Whilst it was not possible to implement all the suggested changes, principally due to contrasting opinions of the reviewers, we have endeavoured to implement as many as possible in a logical manner. Our responses (in

10   red) to specific reviewer comments are found below.

**Reviewer 1 (RC1) J.W. Atkins (Referee)**

**Specific comments**

Pg 1, line 26 need an "of" between "amounts" and "carbon"

15   Response: amended.

Pg 2, line 4 – this sentence is worded awkwardly and could be focused more. Monitoring would imply fixed-chambers, but many studies employ portable chambers some of which require in-situ collars and some that don't.

Response: altered 'monitored' to 'employed for sampling'.

Pg 2, line 13 – "these studies have focussed on a single vegetation type or land use thus do not resolve . . ." Focused has an

20   "s" too many and you need a conjunction between "use" and "thus."

Response: We recognise that US English may prefer 'focused' but according to the journal's guidelines, we have consistently used UK English our manuscript, hence 'focussed'. Addition of comma instead of conjunction.

Pg 2, lines 10-20 – There is some work from Diego Riveros-Iregui that would be a valuable contribution here about diurnal hysteresis if not in this section to set the scene, perhaps later: Riveros-Iregui, D. A., Emanuel, R. E., Muth, D. J., McGlynn,

25   B. L., Epstein, H. E., Welsch, D. L., ... & Wraith, J. M. (2007). Diurnal hysteresis between soil $CO_2$ and soil temperature is controlled by soil water content. Geophysical Research Letters, 34(17).

And this paper may also be useful:

Ruehr, N. K., Knohl, A., & Buchmann, N. (2010). Environmental variables controlling soil respiration on diurnal, seasonal and annual time-scales in a mixed mountain forest in Switzerland. Biogeochemistry, 98(1-3), 153-170.

30   Response: We appreciate being pointed to these relevant studies and refer to them later in the text (p4 L6).

Pg. 3, line 30 – just a note on units grams per hour or micromoles per second are typically more common in the literature.

Response: We recognise that units of mass can cause confusion and have therefore made a clarification: all fluxes are now explicitly expressed in (g $CO_2$ mg m$^{-2}$ h$^{-1}$)

Pg. 4, lines 1-3 – I don't completely understand what you are saying with this phrase," . . . and daily means at 09.00 and ca. 20.00 for all three months in barley" could you clarify that? Are you saying that is when fluxes approximate daily means? And a really, really minor point, but I think "greater" works better to describe fluxes than "higher" because you are talking about a magnitude, an accumulating sum of sorts.

Response: Reworded sentence for clarity "For all three months in barley, maximum $R_s$ was seen between 12:00-15:00, minimum around 05:00 and daily means at 09.00 and *ca.* 20.00.". Amended 'higher' to 'greater'.

Pg. 3, lines 16-20 – Love it. That is a great point and I am enthused to see this work on experimental design and sampling! That is a good highlight to show that difference and make that point about missing differences between the systems.

Response: We are glad that this is appreciated!

Pg. 4. – It would be helpful if you showed your soil moisture data or described it in some way and provide analysis of how that is working with temperature or in isolation to control fluxes. That interaction can be important. There are various ways to look at the interaction of temperature and moisture such as an ANCOVA or even looking at some log regression detrending. Inclusion of an ANCOVA would likely address this and be of minimal additional work.

Response: We reference multiple regression analyses which show that soil moisture is not important on the diurnal scale "Inclusion of soil moisture in a multiple regression did not improve the model, indicating that soil moisture does not affect $R_s$ on the diurnal scale".

Fig. 1 – there is a bit of an over-plotting issue with the data that could be addressed by perhaps widening the plot or decreasing the marker size.

Response: The figure has been amended to include a separate panel for each crop for clarity.

Fig. 2 – Great plot in general, but I think that changing the scale on the y-axes, though I understand visually why it was done, is not a good practice. Normalizing those axes would also better show monthly differences as you can see in the soil temp. plots at the bottom.

Response: We have amended the figure so all the y axes are consistent.

**Reviewer 2 (RC2) Anonymous referee #1**

**Specific comments:**

P. 1, l. 8-10: I don't follow the logic of the sentence. The first half refers to a mode of measurement, based on "convenience" of working in daylight hours, the second invokes an assumption that temperature is a dominant control of soil $CO_2$ efflux. Why this conflation? I assume you want to set up the issue of contrasting diurnal maxima periods, but this is not at all clear in the way it is phrased.

Response: Amended the sentence, replaced "with" with "convenience which is justified by". Our aim here is not to conflate two issues but illustrate how a tacit assumption that diurnal variation in $R_s$ will be controlled by soil temperature (as we show

through the literature cited), and the further assumption that this is consistent at a single location (as shown in our data), has encouraged a sampling convention that synchronous measurements will facilitate valid comparisons to be made, which might not be the case.

P. 1, l. 12: The statement that $R_s$ in Miscanthus peaked in the night is not true. For May and June, Miscanthus "peaks" during the 9:00 – 16:00 window. Diurnal variations in July are subtle (+/- 10%), and interpretation should take account of this magnitude.

Response: This sentence has been re-worded: "whereas in *Miscanthus* after an initial early evening decline, $R_s$ increased above the daily average during the night and in July maximum daily rates of $R_s$ were seen at 22.00".

P. 2, l. 3-7: Here is the same conflation of measurement mode and temperature control. The two concepts are not logically linked here – the single measurement is not a consequence of temperature being widely held as a dominant control on respiration, as the sentence suggests.

Response: An amendment has been made to provide further clarity and to avoid conflation: "and if combined with an assumption that soil temperature will be consistent across a single site, a logical expectation might be that the diurnal variation in $R_s$ will also be consistent at that site".

p. 3, l. 30: As you chose to express fluxes on a mass basis, please specify whether these are grams of carbon, or grams of $CO_2$. Using molar units would avoid any confusion.

Response: For clarity we have amended flux units to mg $CO_2$.

P. 4, l. 12: "fully"???

Response: Now "as much as".

P. 4, l. 17: Which protocol do you refer to here?

Response: Re-worded the sentence for clarity "if a protocol which used the same sampling hour were used over several months".

P. 4, l. 30-32: Picking-and-choosing your data points so they fit the narrative is not appropriate. The temperature response for both data series have to be balanced, and you should show all hourly data for Miscanthus in Fig. 4. Or, as you are interested in a temperature regression across all months, I'm not sure that hourly data are meaningful to show in any case. It's a shame that you don't have temperature data for June in Miscanthus, but for a seasonal temperature response (which is what is sown by regression lines in Fig. 4), you can use monthly average Rs and soil temperature measured in barley as an approximation. From Fig. 2, this would place fluxes of around 300 mg CO2 m-2 h-1 near 12 deg C – what does that do to your curve? Regarding your regression functions – is an apparent saturation curve for the temperature response meaningful for Miscanthus? Finally, there seems to be a mismatch between short-term temperature response (e.g. June in barley, where diurnal flux response to temperature change is very sluggish) vs. seasonal response – this may be worth commenting on.

Response: We acknowledge that our analysis might be perceived as subjective, therefore we show a balanced comparison of all hours for both crops and for the hours of the measurement window in an amended Fig. 4.

P. 5, l. 5: delete "however" (not needed as you start the sentence with "although")

Response: Deleted

P. 4, l. 32 – p. 5, l. 2: I don't completely follow this analysis. Why do you suppose that the relationship between solar radiation and soil CO2 flux is linked to the "typical measuring window"? It seems an entirely arbitrary separation of daytime/nighttime of your data set. What I can see in these graphs is that by introducing a time shift between two essentially sinusoidal curves, you can create an apparent correlation. . The same analysis would work for soil temperature with a time lag, but I obviously see what you're getting at with the lag analysis. An analysis of regression between instantaneous flux and preceding photosynthesis (or radiation used as a proxy) would be more meaningful. If what you try to show is the case, then the deviation from the mean in CO2 flux should be greater during nights following days with high radiation, and less following days of low photosynthesis (i.e. low radiation).

Response: The regression of $R_s$ and time-lagged solar radiation is included in Fig. 5.

Figure 1: Placing both data series on top of each other is not helpful. Please split into separate panels. What happened around the 20th July in the barley field? It seems strange that fluxes should suddenly fall dramatically and then remain constant for days (with only little diurnal variation visible), to then jump back. Any hints in the meteorological data or management (harvest)? What is the impact on your diurnal calculations?

Response: Fig. 1 amended to show two panels. Soil moisture dropped below 0.16 $m^3$ $m^{-3}$ for the only period of the study. This is now referenced (P5 L11): "however, after two weeks without rain, soil moisture dropped to a low of 0.16 m3 m-3 for a short period (19th – 22nd July) in the arable crop, during which time Rs dropped considerably. When heavy rainfall elevated soil moisture rates of Rs increased again which would suggest there is a threshold above which soil moisture is not limiting, an effect similar to that described by Xu and Qi (2001)".

Figure 3: I'm not sure that this graph provides much new information It should be the same as Fig. 2, only that average fluxes per hour and month are multiplied by the number of measurement days, or not? Dynamics should hence be identical.

Response: Whilst we acknowledge the reviewer's observation that the hourly dynamics of cumulative flux are the same as the diurnal variation in $R_s$ values, we feel it is appropriate and useful to show the additional information of the cumulative flux when all hours are integrated, *viz.* the final columns of Fig. 3.

**Reviewer 3 (RC 3) Anonymous referee #4**

p1L14: "coincided with levels" - unclear, reword (see also comment on p5L32).

Response: This sentence has been reworded "Since the time of the daily mean $R_s$ in *Miscanthus* occurred when $R_s$ in the barley was 40% greater than the daily mean".

2.1 This section in general: How often and for how long were the chambers closed?

Response: We have moved a sentence from section 2.2 to 2.1 (P2 L32) for clarity: "The chambers were programmed to close for two minutes during measurement, with a 30 second 'dead band' to allow for mixing of the headspace in a continuous cycle between chambers".

p2L23: ...and *an* infrared gas analyser? p2L24: specify: was it 2 multiplexers (one per ecosystem?) p2L28: inserted 2 cm: It is not mentioned which collar height was chosen (Li-Cor's standard?) and/or how high they protruded above the soil surface. In general, an insertion depth of 2 cm is rather low (possible lateral diffusion in coarse soils) and the resulting large height above the surface should be avoided because of its altering effect on insolation, precipitation and wind (probably not so much an issue once the plant canopy is closed).

Response: Additional clause included (P2 L28): "with one IRGA and one multiplexer deployed in each crop". Whilst we acknowledge that there are effects of collar height, we ensured that these effects would be consistent between the two crops and agree with the reviewer that this was less of an issue since we were measuring under a canopy. We therefore prioritised concerns regarding cutting fine roots over the effects highlighted by the reviewer, and have included an additional reference (Heinemeyer et al.) to explain our reasoning- see methods section (P2 L31): "chambers were seated over PVC collars (diameter 20 cm, height 10 cm) which were inserted ca. 2 cm into the soil in order to minimise the effect of cutting fine roots (Heinemeyer et al., 2011)".

p2L30: Give more details on sensor installation (vertical or through a trench, resulting depth averaging). Note that to gain confidence in the later discussion on (partly lagged) responses to temperature and solar radiation, the temperature would ideally have been measured in several depths.

Response: Additional information "using vertically-installed sensors".

2.2 p3L6: Licor software and manual sounds a bit odd, maybe "manufacturer"?

Response: Re-worded: "using the manufacturer's software (see manufacturer's manual https://www.licor.com/documents/jtpq4vg358reu4c8r4id.pdf)".

p3L14:duplicate dot after 80%

Response: Deleted

p3L17-20: Try to secure the reproducability of the statistical methods not so much (or at least not only) by telling which option of the applied software was chosen, but rather by referring to the name of the test, to literature if necessary, etc., e.g. which test for normality? The result on normality does not seem to be mentioned in the results section (if I didn't overlook it). Note that for soil respiration in general it wouldn't be surprising if it was lognormal rather than normal, where necessary some authors work with log-transformed values.

Response: A clarification of statistical approach has been added to the methods with an appropriate citation (P3 L20): "The cumulative fluxes for the whole period were tested for normality using a Kolmogorov-Smirnov (K-S) test, but due to the size of the dataset this approach was unsuitable for the cumulative fluxes for sampling hour and instead limits of kurtosis and skewness of $\pm$ 2 were used as acceptable deviation from a normal distribution (Field, 2013). Differences in the whole-period cumulative flux were tested using one-way analysis of variance; the effect of crop, sampling hour and month were tested using a mixed-effects model accounting for the repeated estimated totals from each chamber for each month (PROC MIXED in SAS, using the 'repeated' statement and an autoregressive covariance structure)."

P4 L17: The results of tests for normality are reported: "The data did not significantly differ from a normal distribution (K-S test $D_{[10]}= 0.21$, $p> 0.05$; kurtosis= 0.25, skewness= 0.95).".

3.1 p4L3: 9:00 and 20:00: unclear, you mean that instantaneous values close to the daily mean were reached at these times of the day? Reword.

Response: Have reworded the sentence, as was also highlighted by RC1 above. "Reworded sentence for clarity "For all three months in barley, maximum $R_s$ was seen between 12:00-15:00, minimum around 05:00 and daily means at 09.00 and *ca.* 20.00."".

p4L18: "...shows that the shift [...] would be totally missed": This type of very straight conclusion would better fit in the following paragraph, where such things are plainly demonstrated.

Response: Have moved this sentence to P4 L20:

"There was a significant interaction between sampling hour and crop type ($F_{[23,568]}= 3.40$, $p< 0.0001$), and a further significant interaction between crop and month ($F_{[2,568]}= 202.44$, $p< 0.0001$), emphasising that it is not at all valid to assume that measurements made in the adjacent two crops at the same time were sufficient for comparisons of total $R_s$ flux.

Questions must be raised regarding the validity of using blanket, common sampling strategies to compare $R_s$ between different vegetation types, given the marked diurnal changes in $R_s$ demonstrated here. Indeed, if a protocol were employed which used the same sampling hour were used over several months, the significant interaction between shows that the shift from higher $R_s$ in the *Miscanthus* in May to higher fluxes from the barley in June and July would be totally missed."

3.2 p5L1/Fig.5: Make clear that the lag shown in the figure for each months is the one that yielded the optimal $R^2$ after experimentally testing all lag times in a range from x to y in steps of z (here and/or near p3L25 in 2.2).

Response: Amended the caption: "and the lag times shown for each month are those which yield the closest relationship (highest $R^2$)".

p5L3-7: Although this hypothesis is plausible for your case, little is presented to support or falsify it. If radiation data are experimentally shifted to improve $R^2$, so should be temperature data to check for the effect of the mentioned lagged response by improper temperature measurement depth (ideally it would have been measured at more than 1 depth, see comment on p2L30). The physically most consistent way to do so would be by Fourier analysis, since heat transport in the soil would introduce different delays for temperature variations on different temporal scales (e.g. diurnal cycle vs. slower or faster variations), but if variability in a certain time-window is strongly dominated by the diurnal cycle, a simple shifting might do as well. Also, the sentence is very long. Its 2nd half is unclear to me and should be reworded. It seems that a single case study, where hysteresis in the Rs-T relation could be attributed to photosynthates after comprehensive measurements, is used to infer that the same is true in your case. At the same time, an abundance of literature is ignored which demonstrates that also heat transport and measurement depth effects alone can cause hysteresis (e.g. Pavelka et al., 2007, Plant Soil 292:171 and Graf et al., 2008, Biogeosciences 5:1175 to mention just the earliest systematic studies, many follow-ups have been already mentioned by other reviewers).

Response: We refer to the revised Results and Discussion section. These papers are now included amongst other citations (see response to SC4, below) and we feel a balanced interpretation of our data has been presented. We acknowledge several papers (Pavelka et al. 2007; Graf et al. 2008; Oikawa et al 2014; Ruehr et al. 2010; Riveros-Iregui et al. 2007, Philips et al. 2011) which discuss hysteresis of $R_s$ and soil temperature, but we also provide several examples of lagged $R_s$ which suggest an effect of photosynthate (Xu and Qi 2001; Valdocchi et al. 2006; Gavrichkova and Kuzyakov 2008; Kuzyakov and Cheng 2004; Heinemeyer et al. 2011; Barron-Gafford et al. 2011; Zhang et al. 2015). We further reiterate that while we propose an explanation for the observations presented here, a definitive explanation was beyond the remit or expectation of this study (P6 L3).

4 p5L32: Specify what exactly (e.g. the ratio or difference in total repiration between two treatments) can be incorrect by 40 % - the way it is written now suggests that conclusions are, but what would be a 40 % incorrect conclusion?

Amended this sentence: "by as much as 40% relative to the respective daily means".

**Short comment #4 (SC4) Q. Zhang**

The paper is a nice work for guiding soil respiration measurement design. Since the temperature response of soil respiration is so important to your topic, I assume the widely reported soil respiration-temperature hysteresis should be addressed. And you did discuss a little in 3.2 Environmental control of Rs, however, I think this section could be discussed even better by incorporating the knowledge from a few previous efforts. Please see follows. For the diurnal scale soil respiration-temperature hysteresis, there are a few representative works, including the classic Phillips et al. (2011) paper that applies mathematical models answering a few fundamental questions, like how soil temperature measurement depth selection, heat flow influence the respiration-temperature relation, etc; Afterwards, Zhang et al. (2015) combined both model exercise and field experiments to give a more comprehensive explanation of the occurrence and mechanism of the hysteresis. To exclude the possible effect of temperature depth selection by plotting respiration and temperature colocated at the same depth, this work demonstrated how heat flow, gas diffusion, photosynthesis contribute to the hysteresis, and also explained how soil moisture modulates hysteresis magnitude. Actually, the hysteresis may be more widely reported than the authors realized, see the literature list that reported field measured soil respiration-temperature hysteresis in Zhang et al. (2015). As a useful knowledge to this manuscript, the argument that "Even the CO2 flux (F(z)) and the environmental conditions at the same depth can be out of phase, since the flux integrates sources from other depths, causing hysteretic loops" (Zhang et al., 2015) would help explain why the temperature-depth selection cannot avoid hysteresis. Another useful information for this manuscript is related to photosynthesis control on soil respiration. As photosynthesis has long been suggested as the determinant of soil respiration by providing respiration substrate (e.g., Kuzykov and Cheng, 2001; Kuzyakov and Gavrichkova, 2010), Zhang et al. (2015) suggested the time-delayed photosynthesis impact on soil respiration contribute to the '8' shaped soil respiration temperature hysteresis, and altered the hysteresis direction (clockwise cycle, or counterclockwise cycle) under different time lag levels of transferring photosynthate from leaves to roots. But these are numerical modeling representations, Zhang et al. (2015) also acknowledge more field validation are still required. The

authors can think a little about this. A third representative and nice work is by Oikawa et al. (2014) as you cited. references:

Kuzyakov, Y., and W. Cheng (2001), Photosynthesis controls of rhizosphere respiration and organic matter decomposition, Soil Biol. Biochem., 33(14), 1915–1925.

Kuzyakov, Y., and O. Gavrichkova (2010), REVIEW: Time lag between photosynthesis and carbon dioxide efflux from soil: a review of mechanisms and controls, Global Change Biology, 16, 3386-3406.

Oikawa, P. Y., D. A. Grantz, A. Chatterjee, J. E. Eberwein, L. A. Allsman, and G. D. Jenerette (2014), Unifying soil respiration pulses, inhibition, and temperature hysteresis through dynamics of labile soil carbon and O2, J. Geophys. Res. Biogeosci., 119, 521–536.

Phillips, C. L., N. Nickerson, D. Risk, and B. J. Bond (2011), Interpreting diel hysteresis between soil respiration and temperature, Global Change Biol., 17, 515–527.

Zhang, Q., G. G. Katul, R. Oren, E. Daly, S. Manzoni, and D. W. Yang (2015), The hysteresis response of soil CO2 concentration and soil respiration to soil temperature, Journal of Geophysical Research-Biogeosciences, 120, 1605-1618.

We thank the reviewer for the additional references and we have incorporated them into a revised Results and Discussion section (P5 L4- P6 L5).

[revised manuscript text omitted]